# *Rahnella aquatilis* JZ-GX1 Alleviate Salt Stress in *Cinnamomum camphora* by Regulating Oxidative Metabolism and Ion Homeostasis

**Pu-Sheng Li** [1,2], **Wei-Liang Kong** [1,2] and **Xiao-Qin Wu** [1,2,*]

1    Co-Innovation Center for Sustainable Forestry in Southern China, College of Forestry, Nanjing Forestry University, Nanjing 210037, China
2    Jiangsu Key Laboratory for Prevention and Management of Invasive Species, Nanjing Forestry University, Nanjing 210037, China
*    Correspondence: xqwu@njfu.edu.cn; Tel./Fax: +86-25-8542-7427

**Abstract:** Salt stress is an environmental stress that severely limits plant growth, development and productivity. The use of symbiotic relationships with beneficial microorganisms provides an efficient, cost-effective and environmentally friendly preventative method. The plant growth-promoting rhizobacteria (PGPR) *Rahnella aquatilis* JZ-GX1 is a moderately salinophilic strain with good probiotic properties, although its ability to improve woody plant salt tolerance has not been reported. In this study, the effect of JZ-GX1 on *Cinnamomum camphora* under different salt concentrations (0, 50 and 100 mM NaCl) was investigated to reveal the mechanism by which JZ-GX1 improves salt tolerance in *C. camphora*. The results showed that JZ-GX1 promoted plant growth and root development. The relative electrolyte leakage (REL) and malondialdehyde (MDA) production of inoculated *C. camphora* plants were reduced by 37.38% and 21.90%, respectively, and the superoxide dismutase (SOD) activity in the leaves was enhanced by 321.57% under a 100 mM NaCl treatment. It was observed by transmission electron microscopy that under 100 mM salt concentration conditions, the inoculated *C. camphora* leaf cells showed a significant reduction in plasma membrane–cell wall separation and intact chloroplast structures, with tightly packed thylakoids. Importantly, inoculation reduced $Na^+$ accumulation and promoted $K^+$ accumulation in the seedlings, and these changes were consistent with the upregulated expression of the $K^+$ channel *SKOR* and the vesicular membrane $(Na^+, K^+)/H^+$ reverse cotransporter *NHX1* in the plant roots. This study revealed the mechanism of the *Rahnella aquatilis* JZ-GX1 enhancing salt tolerance of *C. camphora*.

**Keywords:** plant growth-promoting rhizobacteria; *Rahnella aquatilis*; salinity tolerance; *Cinnamomum camphora*

## 1. Introduction

Salt stress is one of the major abiotic stresses causing physiological, molecular and biological changes in plants [1]. According to statistics, saline soils are found in more than 100 countries around the world, covering a total area of 1 billion $hm^2$ and accounting for 10% of the global land area. Moreover, saline land is growing worldwide at a rate of $1.0 \times 10^6$ to $1.5 \times 10^6$ ha per year [2]. High concentrations of sodium ions in the soil can inhibit nutrient uptake and photosynthesis in plants, causing severe disruptions in plant metabolism and phenotype [3]. Many methods have been developed to mitigate the severe effects of salt stress on plants. The use of plant growth-promoting rhizobacteria (PGPR) has been highlighted as a promising broad-spectrum method for improving plant growth [4,5]. Methods that use PGPR to improve salt tolerance have significant advantages compared to other methods [6]. They are economical, environmentally friendly and low cost [7].

Research on PGPR to help plants resist salt stress is currently focused on herbaceous plants, including *Hordeum vulgare* inoculated with *Bacillus mojavensis* S1 and *Pseudomonas*

*fluorescens* S3 [8], *Solanum lycopersicum* inoculated with *Pseudomonas* sp. UW4 [9], *Panax ginseng* inoculated with *Paenibacillus yonginensis* DCY84T [10], *Helianthus annuus* inoculated with *Pseudomonas aeruginosa* PF23 [11], *Brassicanapus* inoculated with *Brevibacterium iodinum* RS16, *Micrococcus yunnanensis* RS222 and *B. aryabhattai* RS341 [12], *Arabidopsis thaliana* inoculated with *Paraburkholderia phytofirmans* PsJN [13], *Arachis hypogaea* inoculated with *Klebsiella*, *Pseudomonas*, *Agrobacterium* and *Ochrobactrum* [14], *Zea mays* inoculated with *B. amyloliquefaciens* SQR9 [15], *Mentha arvensis* inoculated with *Halomonas desiderata* STR8 and *Exiguobacterium oxidotolerans* STR36 [16], *Cicer arietinum* inoculated with *Planococcus rifietoensis* RT4 and *H. variabilis* HT1 [17]. However, there are relatively few studies on the effects of PGPR on salt tolerance in woody plants, so we chose woody plants as the research material in this study.

*Rahnella aquatilis* JZ-GX1 is a PGPR isolated in our laboratory from the roots of *Pinus massoniana* [18]. Previous studies have shown that JZ-GX1 significantly increased the germination rate, fresh weight and primary root length of tomato seeds under salt stress and is a moderately salinophilic strain with excellent growth-promoting functions [19]. However, it is not clear how salt tolerance in woody plants is affected in field soils. *Cinnamomum camphora* is an evergreen broad-leaf tree of *Cinnamomum* of the Lauraceae family and is a common excellent greening tree species in urban and rural areas. *C. camphora* prefers warm and humid climates, with strong sprouting power and well-developed main roots, and is widely planted in southern regions [20]. However, *C. camphora* is not tolerant of saline soils, and the salt content of the soil is required to be within 0.2%. High concentrations of secondary soil salt damage can lead to inhibition or even death of *C. camphora* growth. At present, most studies on camphor salt tolerance have focused on the effects of different salt concentrations and low-temperature stress on the photosynthetic and physiological characteristics of camphor seedlings [21,22]. It should be noted that there are few studies on how to improve the salt tolerance of *C. camphora*. Therefore, the use of PGPR to improve the salt tolerance of *C. camphora* is of great application. The aim of this study was to determine the effect of JZ-GX1 on salt tolerance in *C. camphora* and to investigate the mechanism of salt stress alleviation in *C. camphora* by JZ-GX1 in terms of antioxidant capacity, osmoregulation, ultrastructure, ion homeostasis and gene expression. This study was also conducted as an experimental basis for the *R. aquatilis* JZ-GX1 to improve the salt tolerance and fitness of plants.

## 2. Materials and Methods

### 2.1. Selection of Test Strains and Plant Materials

*R. aquatilis* JZ-GX1 is a PGPR isolated from the rhizosphere soil of 28-year-old *P. massoniana* in Nanning, Guangxi, and is currently stored in the Type Culture Preservation Center of China (CCTCC, No: M 2012439) [18].

Three-month-old *C. camphora* seedlings were acquired from a nursery and planted in 15-cm diameter × 10-cm high plastic pots, with 500 g of soil per pot. Soil samples taken from the inter-rooted topsoil of a forest tree were used as growth medium in the experiment, and the relevant properties were as follows: pH 5.5; organic matter, 11.59 mg/kg; available N, 71.62 mg/kg; available P, 12.57 mg/kg; available K, 133.04 mg/kg and salt content, 0.037%. All *C. camphora* seedlings were cultivated in an open greenhouse. There were 10 pots per treatment, with 2 seedlings in each pot, for a total of 20 seedlings.

### 2.2. Seedling Inoculation and Salt Stress Treatment

The *C. camphora* was treated on 30 March 2021 with a bacterial inoculum of $1 \times 10^7$ CFU/plant. A total of 2 treatments were performed: (1) a single application of JZ-GX1; (2) a blank control CK (no inoculation), and the height and chlorophyll content of the *C. camphora* plants were measured and recorded before the inoculation of JZ-GX1. Half a month after the application of JZ-GX1, salt stress was applied to the seedlings on 15 April at three levels of NaCl: 0 mM, 50 mM and 100 mM. Using the method of massive accumulation and gradual addition of salt, 50 mL salt solution was poured every 5 days within 20 days,

and 200 mL salt solution (including 0.15 g or 0.3 g NaCl) was infused into 50 g soil, that is, the concentration was 0.3% and 0.6%, so the final salt content of stress was 50 mM and 100 mM, respectively, avoiding the death of seedlings caused by too much salt. In addition, to ensure that the salt solution in each nursery pot did not leak, the water that seeps into the tray was reintroduced into the soil. The stress lasted for 30 days after reaching the planned final NaCl concentration. The watering was controlled with the same volume of tap water.

### 2.3. Measurement of Plant Growth Parameters

Ten plants were randomly selected from each treatment (after 30 d of stress) and uprooted without damage, and their plant heights were measured using a straight edge (cm). Leaf chlorophyll content was measured with a SPAD-502 handheld chlorophyll meter (Minolta Camera Co., Ltd., Osaka, Japan) and two true leaves at the top of each plant were selected for determination. The seedling height and chlorophyll growth rates were calculated according to percentage changes in the measured plant traits during the treatment period. The underground and aboveground parts of the plants were dried separately in an oven at 70 °C to a constant weight, and then the dry weights of the plants were recorded. The root-to-crown ratio = the dry weight of the underground part/the aboveground part of the plant.

### 2.4. Root System Conformation Determination

Five plants were randomly selected for each treatment; their roots were placed on a transparent tray, and entangled roots were separated with forceps. The *C. camphora* roots were then scanned using a root scanner (Epson Expression 1680, Nagano-ken, Japan), and the resulting images were analysed using WinRHIZO root analysis software (Winrhizo2003b).

### 2.5. Moisture Condition Detection

The relative water content (RWC) and water saturation deficit (WSD) were determined according to the method of Golombek et al. [23]. After 30 d of the stress period, five plants were randomly selected from each treatment, and fresh samples of approximately 0.15 g of leaves were taken to determine their fresh weight (FW). The samples were then immersed in distilled water for 24 h and dried in an oven at 70 °C to a constant weight, and their saturation weight (TW) and dry weight (DW) were determined. RWC and WSD were calculated using the formulae below:

$$RWC\ (\%) = 100 \times (FW - DW)/(TW - DW) \tag{1}$$

$$WSD\ (\%) = 100 \times (TW - FW)/(TW - DW) \tag{2}$$

### 2.6. Determination of Electrolyte Leakage and Malondialdehyde Content in Leaves

Electrolyte leakage (REL) was determined using the conductivity method with reference to Li et al. [24]. Three replicates of each treatment were used. The malondialdehyde (MDA) content was determined by the thiobarbituric acid method [25], with three replicates set up for each treatment.

### 2.7. Observations of the Ultrastructure of C. camphora Leaves

Samples were prepared according to the method of Pareek et al. [26]. Fixation: Fresh samples were taken, quickly cut to approximately 3 mm$^2$ size, immediately fixed in 4% glutaraldehyde solution, left to sink completely, washed 3 times with phosphate buffer for 20 min each time and then fixed in 2% osmium until fully immersed.

Dehydration, infiltration, embedding and polymerization: The phosphoric acid buffer solution was washed three times for 20 min each time; acetone at concentrations of 30%, 50%, 70% and 90% was used for stepwise dehydration, and acetone at a concentration of 100% was used for dehydration three times for 20 min each time for each stage. Then, the

embedding agent was infiltrated and embedded step by step, and the embedding was separately polymerized overnight in a constant temperature box at 37–45–60 °C.

Ultrathin section and staining: Ultrathin sections with a thickness of 50 nm and with uranium and lead double staining were made using an ultrathin microtome and observed under a transmission electron microscope (JEM-1400Flash, JEOL, Tokyo, Japan).

### 2.8. Evaluation of Enzyme Antioxidants and Compatible Solute Content

A leaf sample (0.1 g) was ground in liquid nitrogen and homogenized in an ice bath using the extract from the kit. The homogenate was centrifuged at $8000 \times g$ for 10 min at 4 °C, and the supernatant was collected on ice for testing. According to the instructions of the commercial kits for superoxide dismutase (SOD), peroxidase (POD), catalase (CAT) and reduced glutathione (GSH) (Suzhou Kemin Biotechnology, Suzhou, China), the contents of SOD, POD, CAT and GSH were determined by the trace method.

After weighing 0.1 g of leaves, three replicates were set up for each treatment, and the contents of plant proline, betaine and alginate were determined according to the kit (Suzhou Kemin Biotechnology, Suzhou, China).

### 2.9. Determination of Nutrient Elements and Ions

The roots of *C. camphora* were pre-treated according to the method of Li et al. [19]. Phosphorus content was determined by referring to the method reported by Li et al. [27]. The contents of Ca, Fe, Mn, Cu, Zn, Na and K in the samples were determined by inductively coupled plasma–mass spectrometry (ICP–MS) (Perkin Elmer Optima 2100 DV, Perkin Elmer, Waltham, MA, USA), and a standard curve was drawn to calculate the sodium and potassium concentration values of the samples. There were three repetitions per process.

### 2.10. Quantitative Real-Time Polymerase Chain Reaction Analysis

Total plant RNA was extracted from the roots using an RNA kit (Beijing Zhuangmeng International Biogene Technology) according to the manufacturer's instructions. cDNA samples were prepared using HiScript II Q Select RT SuperMix for RT–qPCR (CAT: 11202ES08; Yeasen, Shanghai, China). The relative expression levels of *SKOR* and *NHX1* were determined by qPCR with an ABI 7500 (Applied Biosystems, Foster City, CA, USA), and *EF1α* [28] was used as an internal control (Table 1). The relative changes in gene expression were calculated by the $2^{-\Delta\Delta CT}$ method. The qPCR assay consisted of three independent experiments with three replicates of each experiment.

**Table 1.** Primers used in the RT–qPCR analysis.

| Gene Name | Gene Function | Primers |
|---|---|---|
| *EF1α* | Endogenous control, Reference gene | TCCAAGGCACGGTATGAT CCTGAAGAGGGAGACGAA |
| *SKOR* | Potassium channel | AAGCAGGCTTTTGCGACTTG TGTTCAAGATCGTTCGCCCA |
| *NHX1* | Antiporter isoform | CCATCTCGCCTTCGTATGCT CAAACATTGGGCGCATGACA |

### 2.11. Statistical Analysis

The data were subjected to analysis of variance and independent-samples Student's *t*-tests using SPSS 22.0 software, and the mean values plus standard errors and significance levels were calculated. Different letters indicate significant differences between control and JZ-GX1-inoculated treatment groups ($p < 0.05$).

## 3. Results

### 3.1. Growth Promotion of Cinnamomum Camphora by Rahnella Aquatilis JZ-GX1

Table 2 shows that all growth indexes of *C. camphora* seedlings under non-salt stress and salt stress were improved. *C. camphora* seedlings inoculated with JZ-GX1 were significantly

higher than the control by 35.39%, 54.58% and 34.74% at salt concentrations of 0, 50 and 100 mM, respectively. The chlorophyll contents estimated as SPAD value of the inoculated plants were significantly increased under different salt stresses. The seedling height and chlorophyll growth rates decreased with increasing NaCl concentration, indicating that high salt stress inhibited the growth of *C. camphora*, whereas the seedling height and chlorophyll growth rates increased with JZ-GX1 inoculation, indicating that JZ-GX1 alleviated this inhibitory effect.

**Table 2.** Effects of different treatments on the camphor growth under salt stress.

| NaCl (mM) | Treatment | Shoot Height (cm) | Shoot Height Growth Rate (%) | SPAD | SPAD Growth Rate (%) |
|---|---|---|---|---|---|
| 0 | CK | 7.12 ± 0.71 [a] | 35.10 | 28.5 ± 3.05 [a] | −17.65 |
| | JZ-GX1 | 9.64 ± 0.37 [b] | 44.53 | 37.88 ± 1.80 [b] | 14.89 |
| 50 | CK | 5.90 ± 0.72 [a] | 6.31 | 26.52 ± 2.12 [a] | −22.27 |
| | JZ-GX1 | 9.12 ± 1.31 [b] | 51.24 | 33.54 ± 4.07 [b] | 7.16 |
| 100 | CK | 5.70 ± 0.17 [a] | 3.26 | 26.50 ± 4.85 [a] | −24.35 |
| | JZ-GX1 | 7.68 ± 0.33 [b] | 24.27 | 37.12 ± 4.30 [b] | 10.87 |

Data are represented by mean ± standard error, n = 3. Different letters indicate statistically significant differences ($p < 0.05$) among treatments according to the least significant difference test.

In this study, although salt stress reduced the dry weight of the aboveground parts and roots of the saplings, the biomass of the inoculated plants was higher than that of the non-inoculated plants at all NaCl concentrations (Figure 1), indicating that strain JZ-GX1 alleviated the damage caused by salt stress to *C. camphora*.

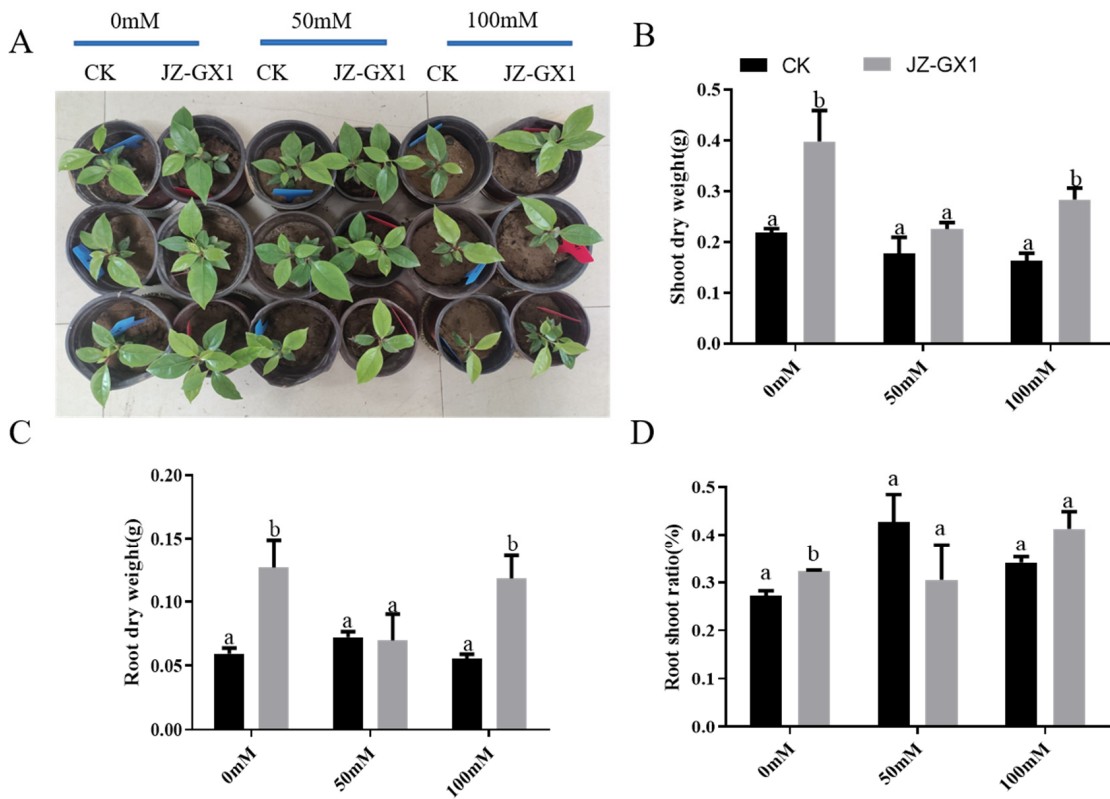

**Figure 1.** Effects of salt stress and inoculation with *Rahnella aquatilis* JZ-GX1 on the biomass of *C. camphora*. (**A**) Phenotype; (**B**) shoot dry weight; (**C**) root dry weight; (**D**) root–shoot ratio. Different lowercase letters indicate significant differences ($p < 0.05$).

Treatment with strain JZ-GX1 induced a significant gain in the dry weight of the aboveground parts of *C. camphora* at 0 mM and 100 mM salt concentrations compared to

the uninoculated treatment, with increases of 115.94% and 113.54%, respectively. There was no significant difference at the 50 mM salt concentration (Figure 1B). The dry weight of the roots of balsam fir inoculated with JZ-GX1 was greater than that of *C. camphora* without inoculation at the same NaCl concentration and was significantly different at 0 mM and 100 mM salt concentrations (Figure 1C). Analysis of the data showed that application of JZ-GX1 had a significant effect on the dry weight of both the aboveground parts and roots and that JZ-GX1 inoculation significantly promoted the root shoot ratio of *C. camphora* under 0 mM salt concentration conditions (Figure 1D).

### 3.2. Changes in Root Parameters of Cinnamomum Camphora Caused by Rahnella Aquatilis JZ-GX1

In this experiment, strain JZ-GX1 increased the total root length, surface area, volume, number of root tips and number of branches in the root system of camphor seedlings at all salt concentrations. The positive aspects of this change are that it improves the uptake of water and nutrients by the roots and has a diluting effect on the salt that enters the roots. The total root length of inoculated and non-inoculated plants and the root surface area of inoculated plants did not change significantly with increasing salt concentration. The total root length of inoculated plants was significantly higher than that of non-inoculated plants at 0 and 100 mM NaCl concentrations (76.85% and 169.74% higher than the latter, respectively). The root surface area was significantly higher at 0, 50 mM NaCl and 100 mM NaCl concentrations (149.20%, 78.09% and 192.65% higher than the latter, respectively) than that of the non-inoculated plants (Figure 2A,B). The mean diameter of the root system did not change significantly between inoculation and salt stress, but the root volume and number of root tips of the inoculated plants were significantly higher than those of non-inoculated plants at the same NaCl concentration, with 272.03%, 128.82% and 192.12% higher root volume and more root tips than non-inoculated plants at 0, 50 and 100 mM NaCl concentrations, respectively (Figure 2C,D). The number of root tips was 93.88%, 60.29% and 87.27% higher than that of non-inoculated plants (Figure 2E). In the absence of salt stress, inoculation had no significant effect on the number of branches in the root systems of the plants, but under salt stress, the number of branches in the root systems of the inoculated plants was significantly higher than that of the non-inoculated plants. The number of branches of inoculated plants was 83.15% and 151.20% higher than that of non-inoculated plants at 50 and 100 mM NaCl concentrations, respectively (Figure 2F).

### 3.3. Rahnella Aquatilis JZ-GX1 Maintains Stability of the Cell Structure and Organelles of Cinnamomum Camphora Leaves

Under non-salt stress, there was no significant difference between the ultrastructure of camphor leaf cell membrane inoculated with JZ-GX1 strain and the control (Figure 3A,B), and the cell membrane was close to the cell wall. Under the concentration of 50 mM and 100 mM salt concentration, the mesophyll cells of the control plants had obvious plasmolysis; however, the plasmolysis degree of *C. camphora* mesophyll cells inoculated with JZ-GX1 strain is lower than that of the control (Figure 3C–F).

### 3.4. Improvement in the Water Status of Cinnamomum Camphora by Rahnella Aquatilis JZ-GX1

In this experiment, although there was no significant change in the RWC of the inoculated camphor seedlings, the WSD of the inoculated plants were all lower than those of the non-inoculated plants (Figure 4A,B). The WSD of inoculated plants were reduced by 80.54%, 30.68% and 36.68% compared to those of non-inoculated plants at 0, 50 and 100 mM NaCl concentrations, respectively.

### 3.5. Enhancement of Membrane Integrity and Osmoregulatory Ability of C. camphora by R. Aquatilis JZ-GX1

Salt stress increased the relative REL of inoculated and non-inoculated plant leaves (Figure 5A). The REL of inoculated plant leaves was significantly lower than that of non-inoculated plant leaves at both 50 and 100 mM NaCl concentrations, 15.01% and 37.38%

lower than the latter, respectively. To study the changes in lipid peroxidation biomarkers after treatment with JZ-GX1, we measured the levels of MDA. The MDA content of inoculated and non-inoculated plant leaves increased under salt stress (Figure 5B). The 100 mM NaCl treatment increased the MDA content of inoculated and non-inoculated plant leaves by 10.05% and 34.27%, respectively, compared to that under non-stressed conditions. Under normal and salt stress conditions, the MDA content of inoculated plant leaves was reduced by approximately 4.71%, 16.98% and 21.90% compared to those of untreated plants, respectively.

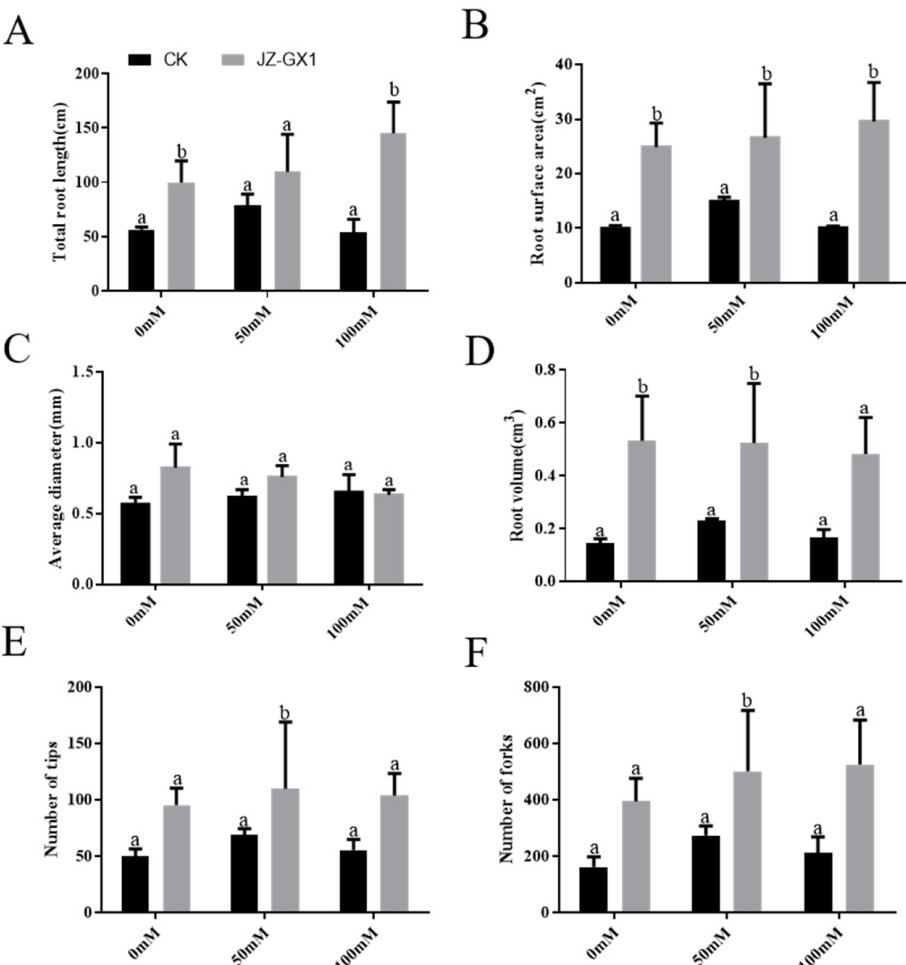

**Figure 2.** Effects of salt stress and inoculation with *Rahnella aquatilis* JZ-GX1 on the morphology of *C. camphora* root systems. (**A**) total root length; (**B**) root surface area; (**C**) average diameter; (**D**) root volume; (**E**) number of tips; (**F**) number of forks. Different lowercase letters indicate significant differences ($p < 0.05$).

Under stress-free conditions, JZ-GX1 inoculation significantly reduced the proline and alginate contents and significantly increased the betaine content of *C. camphora*. None of the three compatible solutes, balsam proline, betaine and alginate, changed significantly under salt stress with JZ-GX1 inoculation (Figure 6A–C).

### 3.6. Enhancement of the Antioxidant Capacity of C. camphora by R. aquatilis JZ-GX1

The antioxidant systems of plants include enzymatic and nonenzymatic systems, of which SOD, POD and CAT are enzymatic antioxidant enzymes, and GSH is a nonenzymatic antioxidant. Under the salt concentration of 50 mM and 100 mM, the activity of SOD in leaves of *C. camphora* was 321.53% and 110.20% higher than that of uninoculated plants, respectively, but the change in roots was not obvious (Figure 7A,B).

In the roots and leaves, plant POD activity did not change significantly with increasing salt concentration, nor did inoculation have a significant effect on plant POD activity (Figure 7C,D). In the roots, salt stress increased the CAT activity of the plants. Inoculation reduced the CAT activity of the plants under stress-free conditions (60.53% lower than the latter). There was no significant effect of inoculation on plant CAT activity under salt stress conditions (Figure 7E,F).

The root GSH activity of the plants decreased continuously with increasing salt concentration, while there was no clear pattern in leaf GSH activity. In the roots, inoculation had no significant effect on the GSH activity of the plants. In leaves, inoculation at 50 mM NaCl reduced plant GSH activity, but there was no significant effect at 0 and 100 mM NaCl concentrations (Figure 7G,H).

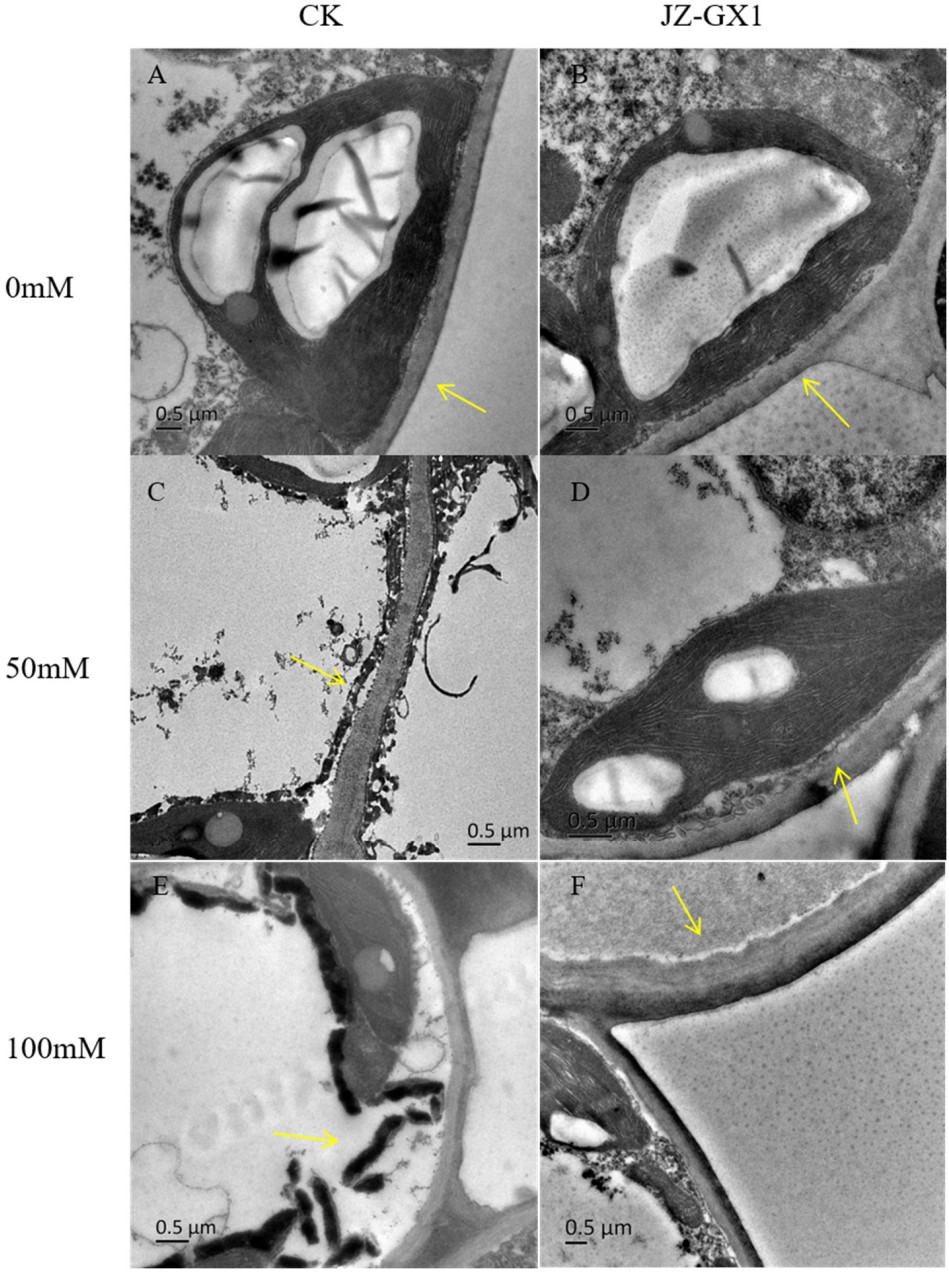

**Figure 3.** Electron microscope images of the cell membrane and cell wall of *Cinnamomum camphora* leaf cells under different salt concentrations. (**A**) 0 mM CK; (**B**) 0 mM JZ-GX1; (**C**) 50 mM CK; (**D**) 50 mM JZ-GX1; (**E**) 100 mM CK; (**F**) 100 mM JZ-GX1. The yellow arrow shows the plasma membrane–cell wall separation.

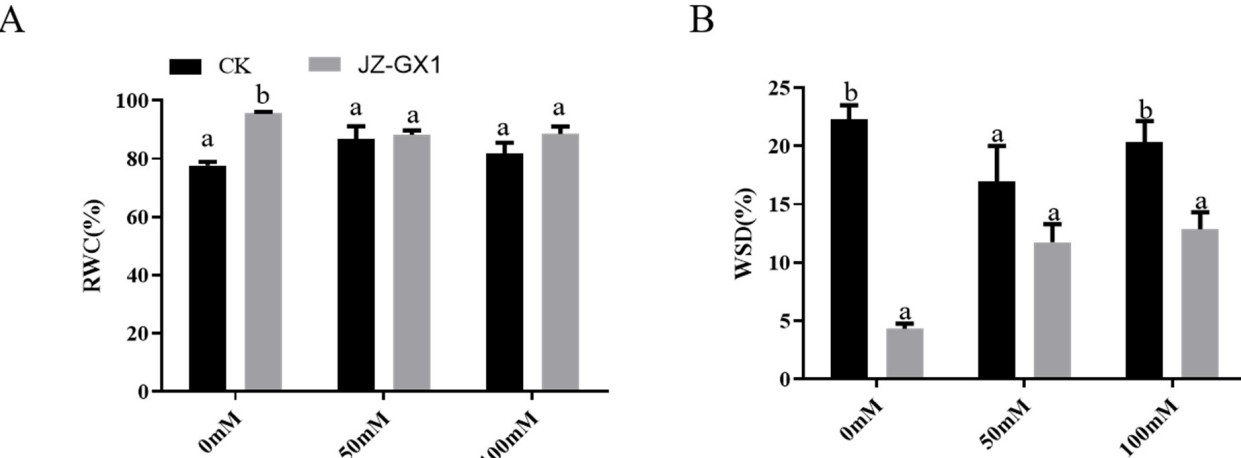

**Figure 4.** Effect of salt stress inoculation with *Rahnella aquatilis* JZ-GX1 on the water status of *C. camphora* leaves. (**A**) Relative water content (RWC); (**B**) water saturation deficit (WSD). Different lowercase letters indicate significant differences ($p < 0.05$).

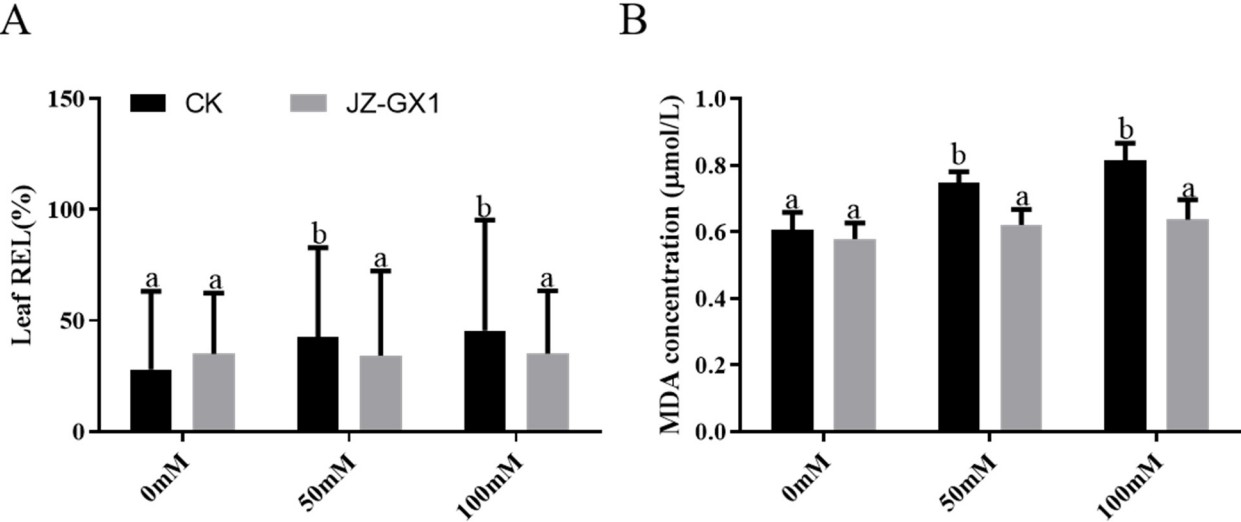

**Figure 5.** Effect of salt stress inoculation with *Rahnella aquatilis* JZ-GX1 on relative electrolyte leakage (REL) and malondialdehyde (MDA) in *C. camphora* leaves. (**A**) Leaf REL; (**B**) MDA concentration. Different lowercase letters indicate significant differences ($p < 0.05$).

*3.7. Nutrient Absorption and Ion Balance of Cinnamomum Camphora Promoted by Rahnella Aquatilis JZ-GX1*

In the absence of salt stress, inoculation reduced the Ca content of the plants, and under salt stress, inoculation had no significant effect on the Ca content of the plant leaves. At low salt concentrations (0 and 50 mM NaCl), inoculation reduced the Fe content of plant leaves, whereas at high salt concentrations (100 mM NaCl), the Fe content of inoculated plants was 22.93% higher than that of uninoculated plants. The Mn content of inoculated plant leaves was reduced by 18.20%, 44.59% and 7.62% under 0, 50 and 100 mM NaCl treatments, respectively, compared to that of uninoculated plant leaves, but the Cu content was increased by 55.40% and 57.28% under 0 and 100 mM NaCl treatments, respectively. Inoculation had no significant effect on the Zn content of the leaves of plants treated with 0 mM NaCl, significantly reduced the Zn content of the leaves of plants treated with 50 mM NaCl and significantly increased the Zn content of the leaves of plants treated with 100 mM NaCl (Figure 8).

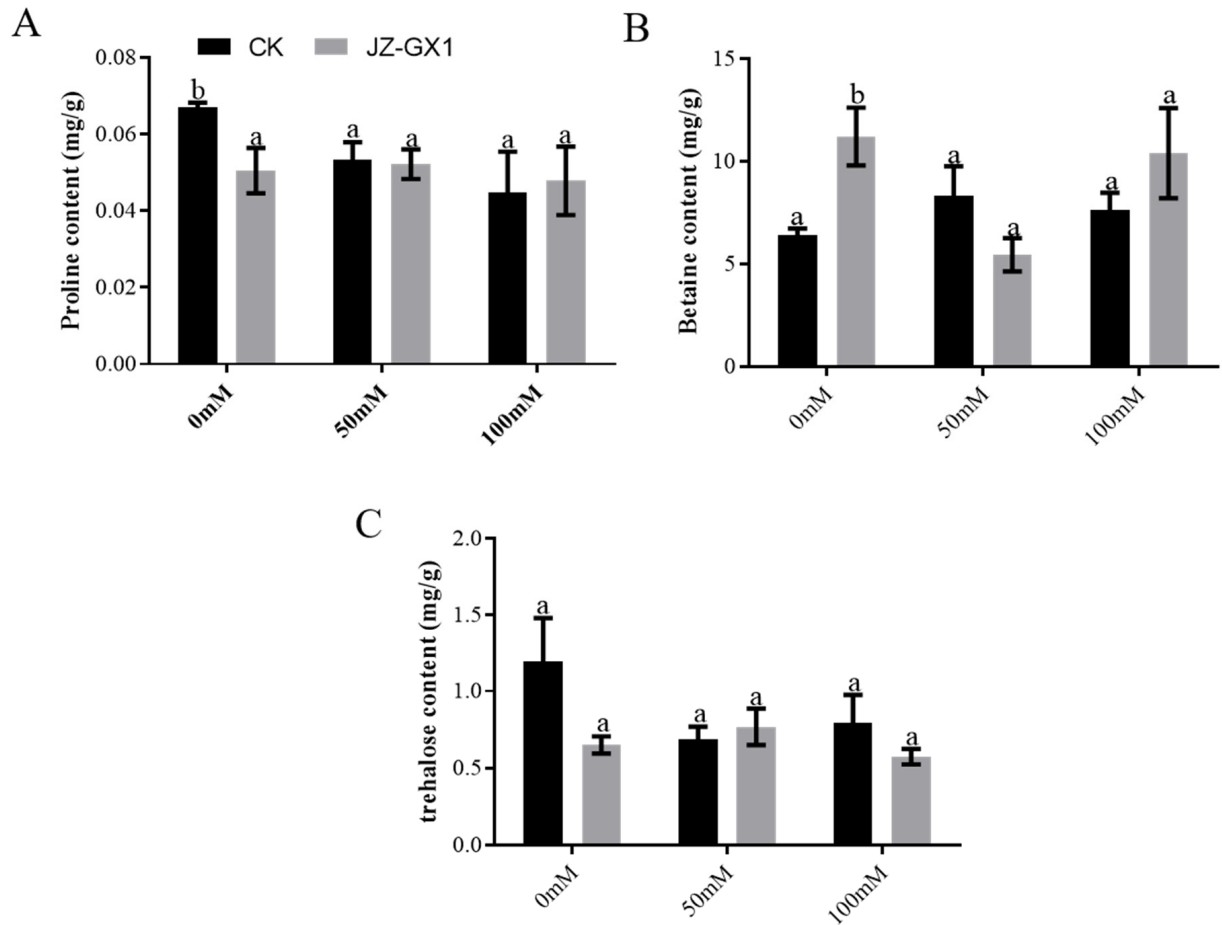

**Figure 6.** Effect of *Rahnella aquatilis* JZ-GX1 on osmoregulatory substances in *C. camphora* under salt stress. (**A**) Proline; (**B**) betaine; (**C**) trehalose. Different lowercase letters indicate significant differences ($p < 0.05$).

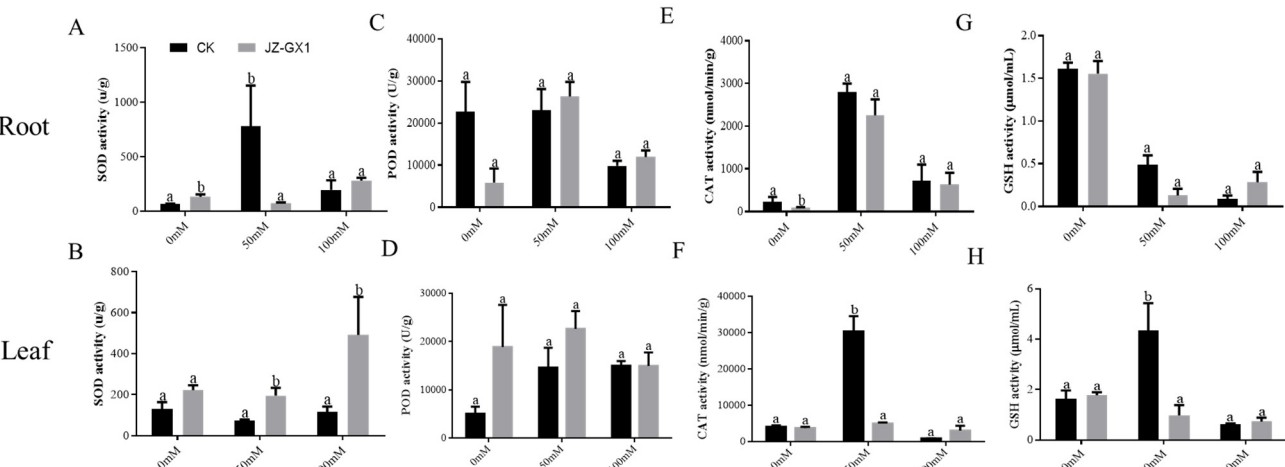

**Figure 7.** Effects of *Rahnella aquatilis* JZ-GX1 on the antioxidant capacity of *C. camphora* under salt stress. (**A**) Root SOD activity; (**B**) leaf SOD activity; (**C**) root POD activity; (**D**) leaf POD activity; (**E**) root CAT activity; (**F**) leaf CAT activity; (**G**) root GSH activity; (**H**) leaf GSH activity. Different lowercase letters indicate significant differences ($p < 0.05$).

Inoculation reduced the $Na^+$ content of *C. camphora* by 17.67%, 7.96% and 11.84% under both salt stress and non-salt stress conditions, respectively. Under non-salt stress conditions, there was no significant difference in $K^+$ accumulation between inoculated and

uninoculated *C. camphora* compared to the control, but inoculation under 100 mM NaCl treatment significantly increased *C. camphora* K⁺ accumulation by 68.26% compared to the latter (Figure 9A,B).

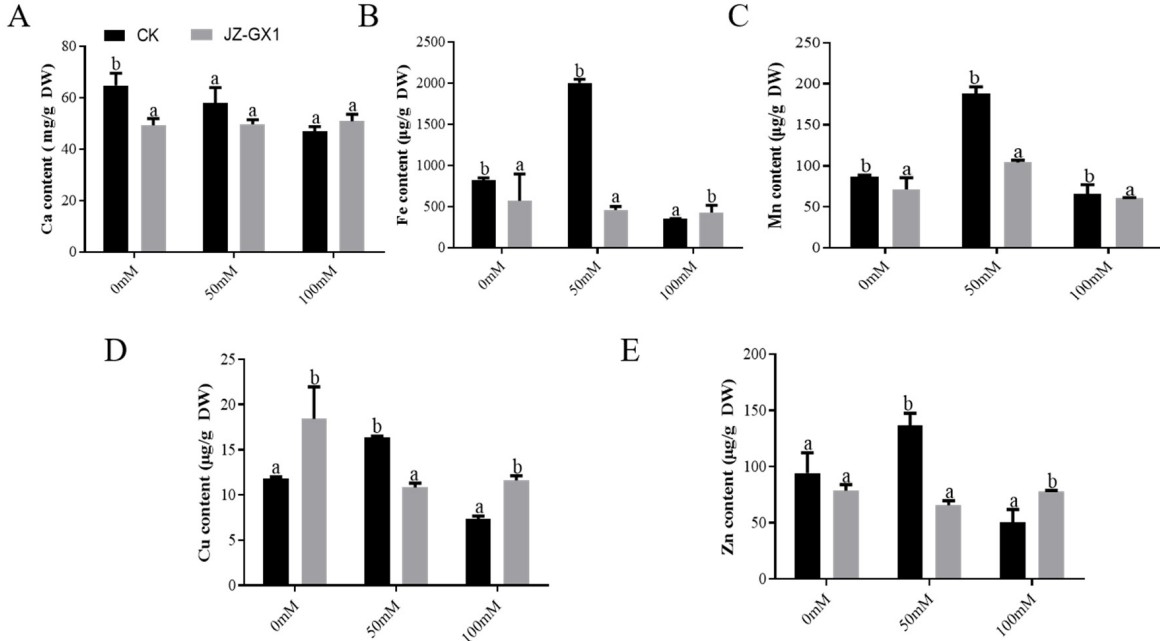

**Figure 8.** Effects of *Rahnella aquatilis* JZ-GX1 on the trace element content of *C. camphora* root systems under salt stress. (**A**) Ca; (**B**) Fe; (**C**) Mn; (**D**) Cu; (**E**) Zn content. Different lowercase letters indicate significant differences ($p < 0.05$).

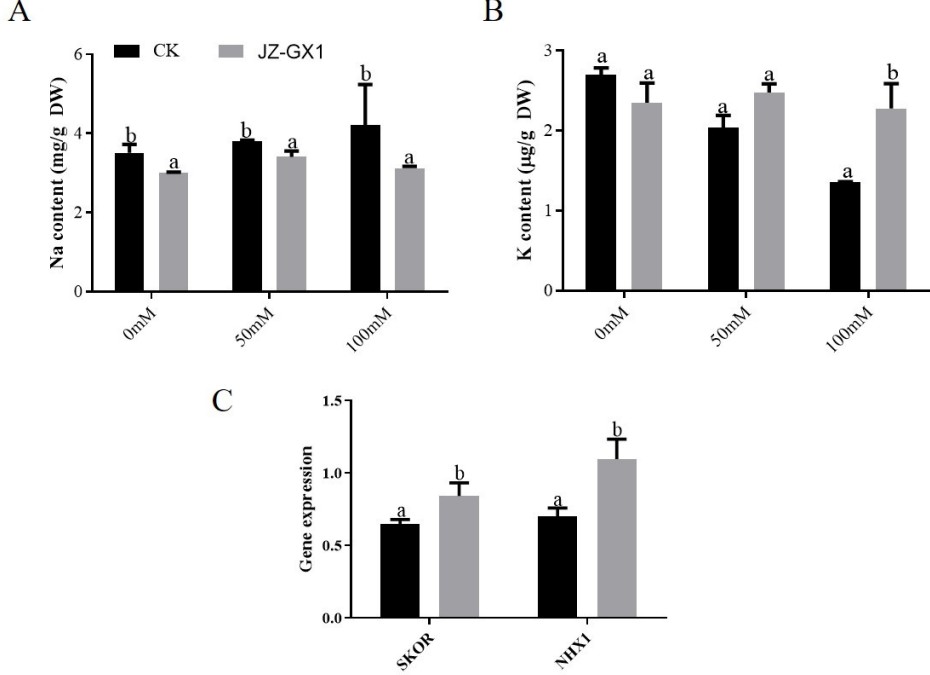

**Figure 9.** Effect of *Rahnella aquatilis* JZ-GX1 on the Na and K contents of *C. camphora* root systems under salt stress. (**A**) Na content; (**B**) K content; (**C**) *SKOR* and *NHX1* expression. Different lowercase letters indicate significant differences ($p < 0.05$).

Because of the positive effects of JZ-GX1 in both reducing root Na and increasing root K under salt stress, we further examined the expression of two genes involved in

K transport, Na efflux and compartmentalization. In the root system, the expression of the *SKOR* and *NHX1* genes was 30.03% and 56.84% higher in inoculated plants than in non-inoculated plants at a concentration of 100 mM NaCl, respectively (Figure 9C).

## 4. Discussion

Salt stress, as one of the major adversity factors governing plant growth and development, significantly reduced the growth parameters of *C. camphora* seedlings. In contrast, the application of the *R. aquatilis* JZ-GX1 effectively alleviated the inhibitory effect of salt stress on the growth of *C. camphora* seedlings, similar to the results of a study conducted by Irizarry and White [29] on cotton (*Gossypium* spp.). Plant biomass best reflects plant performance under abiotic stresses [30]. Inoculation with JZ-GX1 increased plant biomass at all NaCl concentrations. Other studies on tomato (*S. lycopersicum*) [31], rice (*Oryza sativa*) [32], lettuce (*L. sativa*) [33] and maize (*Z. mays*) [34] have also found that the application of PGPR under salt stress can promote plant growth.

The root structure of the plant determines the performance of the root system. A proliferating root system is essential for deeper penetration into the soil layer and easier access to water and nutrients [35]. This study found that inoculation with JZ-GX1 increased the total root length, surface area, mean diameter, volume, root tip number and branching number of *C. camphora* root systems, confirming the positive effect of JZ-GX1 on alleviating salt stress injury in *C. camphora* seedlings. The secretion of indoleacetic acid by JZ-GX1 could also induce root development [19], which ultimately also helps to regulate plant nutrient composition and balance ion influx in plants under salt stress [36]. Changes in root morphology are one of the important mechanisms by which JZ-GX1 enhances the water and nutrient uptake of plants under poor soil conditions.

Salt stress increases the difficulty of water uptake by plant roots and induces osmotic stress in plants, which can adversely affect plant growth [37]. In addition, plants inevitably lose water through transpiration as they photosynthesize. Maintaining a good water status and water balance is therefore beneficial for plant photosynthesis and growth. The results of this experiment showed that strain JZ-GX1 reduced the WSD at 0, 50 and 100 mM NaC1 concentrations and ensured the water balance of the plants. This indicated that PGPR JZ-GX1 promoted plant growth, and this promotion can be attributed to the improvement in the water status of the plants caused by the inoculation treatment. Osmotic stress, triggered by salt stress, can trigger plasma membrane–cell wall separation in plant cells. Therefore, investigating the effect of JZ-GX1 on plant ultrastructure under salt stress will deepen our understanding of the mechanism by which JZ-GX1 improves salt tolerance in plants. Studies have shown that salt stress can damage the ultra-structures of plant leaf cell membranes and chloroplasts. Similar reports of cell wall separation under salt stress have also been reported for *T. foenum-graecum* [38]. The contraction of the cell membranes of the balsam fir plants inoculated with the JZ-GX1 was significantly reduced compared to that of the control, probably due to the higher osmoregulatory capacity of *C. camphora* inoculated with the JZ-GX1. Chloroplasts are structurally intact and ordered to ensure that light energy is converted into photosynthetic products. Salt stress can damage photosynthetic organs and lead to the loosening of vesicle-like membranes [39]. This was confirmed in this study, and JZ-GX1 inoculation allowed the cyst-like bodies to be more compact.

Salt stress induces excessive reactive oxygen species (ROS) production in plant cells, leading to membrane lipid peroxidation and the disruption of cell membrane integrity [40]. MDA is one of the main products of lipid peroxidation, and its level reflects to some extent the degree of membrane lipid peroxidation and the strength of the plant's response to adversity [41]. In this study, 50 mM salt stress increased the MDA content of *C. camphora* by 23.23%, indicating that salt stress caused a large amount of ROS accumulation in *C. camphora* seedling leaves and severely damaged the membrane system, which is consistent with the results of a previous study on wheat [42]. Inoculation with JZ-GX1 reduced the MDA content of the plants, indicating that the degree of oxidative damage to the leaves of the inoculated plants was lower than that of the non-inoculated plants. Under salt stress,

JZ-GX1 inoculation had no significant effect on plant-compatible solutes such as proline, betaine and trehalose. The effects of different PGPRs on plant-compatible solutes can vary. *Bacillus subtilis* SU47 and *Arthrobacter* sp. SU18 can significantly enhance proline in wheat under salt stress, thereby enhancing salt tolerance in wheat [43]. Inoculation with *Azospirillun brasilense* NH reduced the proline content of plants but improved their salt tolerance [44]. JZ-GX1 could inhibit membrane lipid peroxidation to a certain extent, but the compatible solute production was not the mechanism for improving the antioxidant capacities of the plants. In addition to nonenzymatic components, enzymatic antioxidants play a key role in the induction, elimination, detoxification or neutralization of toxic levels of ROS [45]. SOD, POD, CAT and GSH are important protective enzymes of the plant membrane lipid peroxidation enzymatic defence system, and they can effectively scavenge ROS by coordinating with each other and working together [46,47]. JZ-GX1 could increase SOD activity in *C. camphora* leaves, reflecting that the $O_2^-$ removal ability of inoculated plants was higher than that of non-inoculated plants under salt stress, which was beneficial to the quenching of ROS in *C. camphora* seedlings under salt stress and the alleviation of salt stress injury. This is in agreement with the findings of Baltruschat et al. [48] that under salt stress, *Piriformospora indica* enhanced CAT activity and, thus, salt tolerance in barley. It is also worth noting that there was no significant difference in POD activity between inoculated and non-inoculated plants in all treatments, but there was a decrease in GSH activity, suggesting that the increase in salt tolerance by JZ-GX1 inoculation did not indicate an increase in the activity of all antioxidant enzymes and that there was also a decrease in the activity of some antioxidant enzymes. JZ-GX1 can regulate the antioxidant capacity of plants through a combination of enzymatic and nonenzymatic systems to protect against oxidative damage caused by ROS.

Inoculated plants have better growth and higher salt tolerance than non-inoculated plants, which is related to their better nutrient uptake and ionic balance [49]. Phosphorus, an essential nutrient for plant growth, is absorbed by roots in either univariate ($H_2PO_4$) or binary ($HPO_4$) soluble forms. Inoculation with *B. aquimaris* significantly increased the P content of wheat under salt stress conditions [50]. The inoculation of JZ-GX1 in our study did not significantly affect the P content of *C. camphora*. It is noteworthy that the inoculation treatment under 100 mM salt stress increased the levels of a number of nutrients to varying degrees, including the trace elements Fe, Cu and Zn. The accumulation of $Na^+$ in plants destroys various physiological and biochemical processes, such as nutrient balance, water balance, photosynthesis and antioxidant metabolism [51] and disrupts potassium ion uptake by root cells [52]. The results of this study showed that the $Na^+$ content in plants inoculated with the JZ-GX1 was significantly lower than that of the control, indicating that JZ-GX1 promoted $Na^+$ excretion from plant roots and inhibited the uptake of $Na^+$ from the environment by the roots. *B. subtilis* GB03 can also enhance salt tolerance in Arabidopsis, clover and wheat by reducing $Na^+$ content [53–55]. Many physiological processes in plants require the involvement of K, including osmoregulation, protein synthesis, enzyme activation and photosynthesis. $Na^+$ competes with $K^+$ for binding sites in these metabolic processes, but $Na^+$ cannot replace $K^+$ in its bioregulatory roles [56,57]. Increased K content in plants can alleviate the toxic effects of $Na^+$ under salt stress [58]. Therefore, increasing the potassium content is a common strategy for alleviating salt stress. We further examined the K content of *C. camphora* and showed that JZ-GX1 significantly promoted the accumulation of $K^+$ in *C. camphora* roots under 100 mM NaCl treatment. The reason for this may be that JZ-GX1 increases the number of lateral roots of the plant, increasing the contact area and thus effectively improving the ability of *C. camphora* to absorb mineral nutrients from the soil. On the other hand, K is selectively absorbed by the plant as an isotonic substance and transported to the plant organs and tissues, aiding the plant by preventing the absorption of more $Na^+$. In this study, JZ-GX1 treatment increased the content of $K^+$ in plant roots under salt stress. Importantly, JZ-GX1 promoted the upregulation of the $K^+$ channel SKOR and the vesicular membrane ($Na^+$, $K^+$)/$H^+$ reverse cotransporter NHX1 in *C. camphora* roots, contributing to the maintenance of stable ionic homeostasis.

## 5. Conclusions

As mentioned above, this paper proposes a model for the role of the inter-rhizosphere promoting JZ-GX1 in plant salt tolerance (Figure 10). Under salt stress, JZ-GX1 reduced root $Na^+$ content by regulating the expression of related genes, which in turn reduced the salt transfer from roots to leaves. Inoculated plants had higher leaf antioxidant enzyme SOD activity, REL and lower MDA content than uninoculated plants, as well as improved root morphology, reduced water loss, increased chlorophyll content and plant height growth rate, stabilized plant leaf cell ultra-structure and enhanced plant salt tolerance through an affected nutritional status that caused physiological changes in the aboveground parts of the plant. Further understanding whether *Rahnella aquatilis* JZ-GX1 can be used in the field and big trees is essential to facilitate the development of PGPR-mediated abiotic stress amelioration strategies and will provide a sustainable approach for the bioprotection of plants under salt stress conditions.

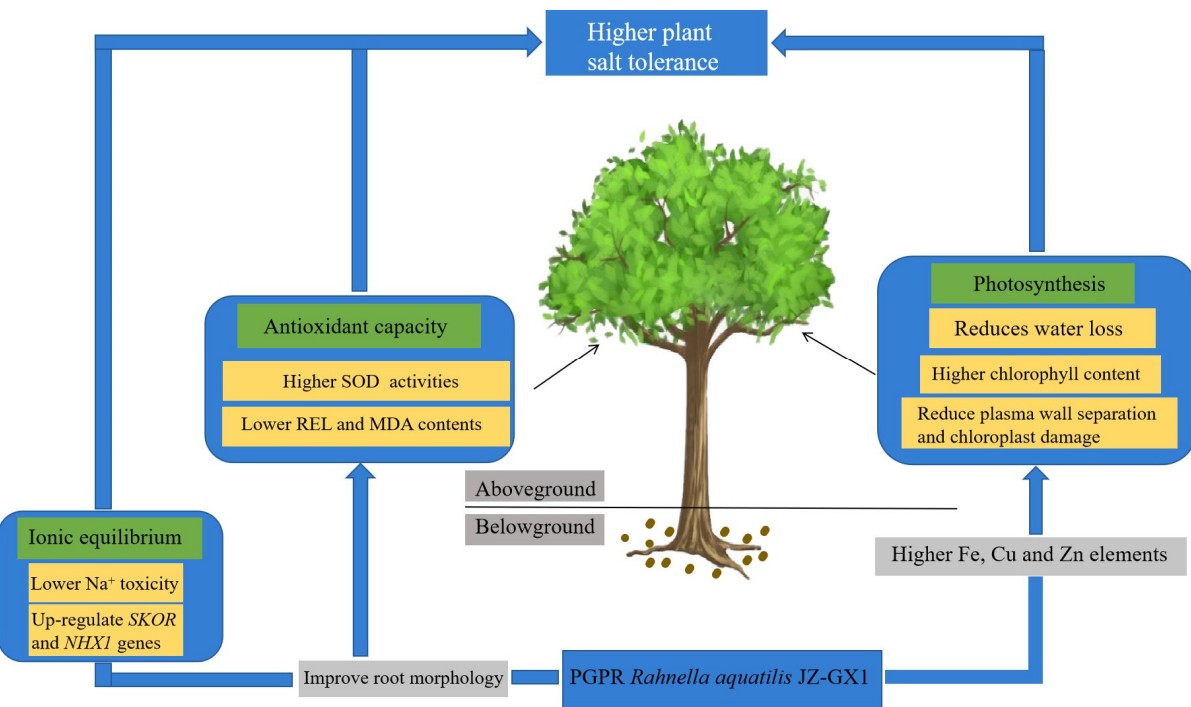

**Figure 10.** Effect of PGPR *Rahnella aquatilis* JZ-GX1 inoculation on *C. camphora* salt tolerance.

**Author Contributions:** P.-S.L. and W.-L.K. completed the experimental research, data analysis and the first draft of the paper; X.-Q.W. directed the experimental design, data analysis, paper writing and revision. All authors have read and agreed to the published version of the manuscript.

**Funding:** This work was supported by the National Key Research and Development Program of China (2017YFD0600104) and the Priority Academic Program Development of the Jiangsu Higher Education Institutions (PAPD).

**Institutional Review Board Statement:** Not applicable.

**Informed Consent Statement:** Informed consent was obtained from all subjects involved in the study.

**Conflicts of Interest:** The authors declare that they have no known competing financial interest or personal relationships that could have appeared to influence the work reported in this paper.

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
