# Peer review of "Rahnella aquatilis JZ-GX1 Alleviate Salt Stress in Cinnamomum camphora by Regulating Oxidative Metabolism and Ion Homeostasis"

_forests, doi:10.3390/f14061110_

Round 1

Reviewer 1 Report

Review on forests-2326329 „Rahnella aquatilis JZ-GX1 Alleviate Salt Stress in Cinnamomum camphora by Regulating Oxidative Metabolism and Ion Homeostasis”

The manuscript is focussing on an environmentally important issue how plant biomass and physiological performance can be improved under salt stress by means of PGPR application. The effect of PGPR on herbaceous plant species is well-studied, but there are relatively little information on woody species. The authors have studied the influence of NaCl on seedlings of Cinnamomum camphora, an evergreen broad-leaf tree species, in controlled experimental conditions using a complex methodology for understanding the effects of PGPR on plant growth and morphology, anatomy at organ and cell levels, biochemical traits and ion homeostasis. Although the topic is interesting and novel I would suggest the Authors to revise the manuscript, especially through making more understandable the description of treatments, measurement protocols and if possible with involvement of results which potentially they have on the ionomics of leaves. I have made questions and comments with signing the lines, which hopefully will help the revision work.

L24 Please correct the expression “plasma wall separation” to e.g. plasma-cell wall separation, or plasma membrane-cell wall separation.

Please also change “with tightly packed cysts” into “with tightly packed thylakoids”.

L36 “Moreover, saline land is growing worldwide at a rate of 1.0×106 to 1.5×106 hm2 per year” – please change the unit of hm2 into a more appropriate one.

L85-88 In 2.2 (Selection of test strains and substrates) the authors should make clear what they consider substrate in the study e.g. soil samples taken from the interrooted topsoil of a forest tree were used as growth medium in the experiment. Then add the description of soil characteristics.

L89-103 In 2.3.(Seedling inoculation and salt stress treatment) the authors should add additional information on the experimental object and growing conditions in order to make reproducibility of results such as:

A detailed description of the experimental conditions is missing: what is the origin of C. camphora seedlings used as experimental plant objects? It is not clear whether seedlings were obtained from seeds in a glasshouse or collected in field, furthermore what age the seedlings have at the start of experiment. How many seedlings they used? What organs were used for measurement of chlorophyll content?

What was the volume of growth medium? They used a mass accumulation gradual salt addition method to reach the planned final salt concentration in different treatments, but it is critical in what growth medium volume they used the salt.

Using this method, the following sentence is not understandable „pouring 50 mL of salt solution every 5 d in 1/4, 1/2 increments to achieve a final stress salt content of 50 mM and 100 mM.”

Concerning the described method, for me it is rather questionable whether the plants were actually treated with the planned salt concentrations (50mM and 100mM NaCl).

L103 „the stress lasted for 30 d”, it is also not clear whether it is related to the whole period of experiment or to that period when the planned final NaCl concentrations were reached. Please correct the description.

L107 Please add precise information on type of the used N-tester (Germany)

L110 Please add precise information on calculation of „the root-to-crown ratio”

L119 Add the proper citation to this „the method of Golombek et al.” 2

L127 In 2.7. (Determination of electrolyte leakage and malondialdehyde content), the authors should add additional information on the organs.

L 165 In 2.11. (Quantitative real-time polymerase chain reaction analysis), the authors should also add additional information on the organs.

L186 I suggest a correction: chlorophyll content estimated as SPAD value, since you use not exact chlorophyll concentration data.

L187-188 how the seedling height and chlorophyll growth rates were calculated – I mean that these rates are only percentage changes in the measured plant traits during the treatment period.

L192 In title of table 2 please add to what the Growth rate(%)values are related to and how they were calculated.

L256-257 Please include this sentence into Discussion section: „ The PGPR JZ-GX1 promoted plant growth, and this promotion can be attributed to the improvement in the water status of the plants caused by the inoculation treatment.”

L262 I suggest a change in the title of 3.5. section since REL and MDA data reflect rather the change in membrane integrity than change in the osmoregulatory ability.

L 286-288 I suggest modification of this sencence and make clear why you think that „glutathione reductase (GSH) is a nonenzymatic antioxidant”.

L293-297 Please make this statement more understandable.

L314-344 In the section 3.7., one would expect how the PGPR influence the concentrations of investigated nutrients in the shoots of the experimental plants which is critical in connection with the improvement of total physiological activity of your plants. If you measured the nutrient in leaves, please include these results in the revised manuscript.

L357-359 I suggest to clarify this statement „Salt stress causes a reduction in the root cortex, shortening the distance between the epidermis and the mid-column, thus facilitating the uptake of essential minerals from the soil [35].”

If this salt stress-related anatomy change appears it would be advantageous due to enhancing effect on the short-distance ion transport into the stele vascular tissue.

L373 Please correct this expression: plasma wall separation in plant cells to e.g. plasma-cell wall separation or plasma membrane-cell wall separation.

L446-448 I do not fully agree with this statement since You measured K only in roots but it is also presumed that decrease in K concentration in the roots might be attributed to the enhanced K transport into the shoots (leaves) under NaCl treatment.

L465 Modelling and better understanding the potential influence of PGPR on the whole plant physiology in salty environment would require more information on the change in ion homeostasis in leaves too.

Author Response

Response to Reviewer 1 Comments

Point 1. L24 Please correct the expression “plasma wall separation” to e.g. plasma-cell wall separation, or plasma membrane-cell wall separation.

Response 1: Thank you very much, the revised details can be found in Line 24-25, page 1.

Point 2. L36 “Moreover, saline land is growing worldwide at a rate of 1.0×106 to 1.5×106 hm2 per year” – please change the unit of hm2 into a more appropriate one.

Response 2: According to your comment, we have revised it. The revised details can be found in Line 38, page 1.

Point 3. L85-88 In 2.2 (Selection of test strains and substrates) the authors should make clear what they consider substrate in the study e.g. soil samples taken from the interrooted topsoil of a forest tree were used as growth medium in the experiment. Then add the description of soil characteristics.

Response 3: Thank you for your careful work. We have made correction according to the Reviewer’s comments. The revised details can be found in Line 89-91, page 2.

Point 4. L89-103 In 2.3.(Seedling inoculation and salt stress treatment) the authors should add additional information on the experimental object and growing conditions in order to make reproducibility of results such as:

A detailed description of the experimental conditions is missing: what is the origin of C. camphora seedlings used as experimental plant objects? It is not clear whether seedlings were obtained from seeds in a glasshouse or collected in field, furthermore what age the seedlings have at the start of experiment. How many seedlings they used? What organs were used for measurement of chlorophyll content?

What was the volume of growth medium? They used a mass accumulation gradual salt addition method to reach the planned final salt concentration in different treatments, but it is critical in what growth medium volume they used the salt.

Using this method, the following sentence is not understandable „pouring 50 mL of salt solution every 5 d in 1/4, 1/2 increments to achieve a final stress salt content of 50 mM and 100 mM.”

Concerning the described method, for me it is rather questionable whether the plants were actually treated with the planned salt concentrations (50mM and 100mM NaCl).

Response 4: We are very sorry for our negligence, according to 1%=170 mM, we convert that 50 mM is about 0.3%, that is, 0.3 g of NaCl is added to 100 g soil, of which 0.3 g of sodium chloride is dissolved in 50 mL distilled water, the concentration of 100 mM, and so on. Two true leaves at the top of each plant were selected for SPAD determination. These revised details can be found in Line 88-96, page 2; Line 105-110, page 3; Line 124, page 3.

Point 5. L103 „the stress lasted for 30 d”, it is also not clear whether it is related to the whole period of experiment or to that period when the planned final NaCl concentrations were reached. Please correct the description.

Response 5: We continued the stress for 30 days after reaching the planned final concentration of sodium chloride. The revised details can be found in Line 116-117, page 3.

Point 6. L107 Please add precise information on type of the used N-tester (Germany)

Response 6: OK. The revised details can be found in Line 122-123, page 3.

Point 7. L110 Please add precise information on calculation of „the root-to-crown ratio”

Response 7: We have made correction according to the Reviewer’s comments. The revised details can be found in Line 128-129, page 3.

Point 8. L119 Add the proper citation to this „the method of Golombek et al.” 2

Response 8: We are very sorry for our negligence, the revised details can be found in Line 138, page 3.

Point 9. L127 In 2.7. (Determination of electrolyte leakage and malondialdehyde content), the authors should add additional information on the organs.

Response 9: We have made correction according to the Reviewer’s comments. The revised details can be found in Line 146, page 3.

Point 10. L 165 In 2.11. (Quantitative real-time polymerase chain reaction analysis), the authors should also add additional information on the organs.

Response 10: We have made correction according to the Reviewer’s comments. The revised details can be found in Line 185, page 4.

Point 11. L186 I suggest a correction: chlorophyll content estimated as SPAD value, since you use not exact chlorophyll concentration data.

Response 11: We have made correction according to the Reviewer’s comments. The revised details can be found in Line 206, page 5.

Point 12. L187-188 how the seedling height and chlorophyll growth rates were calculated – I mean that these rates are only percentage changes in the measured plant traits during the treatment period.

Response 12: We have made correction according to the Reviewer’s comments. The revised details can be found in Line 124-126, page 3.

Point 13. L192 In title of table 2 please add to what the Growth rate(%)values are related to and how they were calculated.

Response 13: We have made correction according to the Reviewer’s comments. The revised details can be found in Table 2.

Point 14. L256-257 Please include this sentence into Discussion section: „ The PGPR JZ-GX1 promoted plant growth, and this promotion can be attributed to the improvement in the water status of the plants caused by the inoculation treatment.”

Response 14: We have made correction according to the Reviewer’s comments. The revised details can be found in Line 396-399, page 13.

Point 15. L262 I suggest a change in the title of 3.5. section since REL and MDA data reflect rather the change in membrane integrity than change in the osmoregulatory ability.

Response 15: We have made correction according to the Reviewer’s comments. The revised details can be found in Line 282, page 9.

Point 16. L286-288 I suggest modification of this sencence and make clear why you think that „glutathione reductase (GSH) is a nonenzymatic antioxidant”.

Response 16: We are very sorry for our negligence, we're talking about reduced glutathione, the revised details can be found in Line 172, page 4; Line 309, page 10.

Point 17. L293-297 Please make this statement more understandable.

Response 17: We have made correction according to the Reviewer’s comments. The revised details can be found in Line 310-312, page 10.

Point 18. L314-344 In the section 3.7., one would expect how the PGPR influence the concentrations of investigated nutrients in the shoots of the experimental plants which is critical in connection with the improvement of total physiological activity of your plants. If you measured the nutrient in leaves, please include these results in the revised manuscript.

Response 18: We understand the concern of the reviewers that we neglected to measure the mineral nutrients in the leaves when designing the experiment, and we will explore it in the future.

Point 19. L357-359 I suggest to clarify this statement „Salt stress causes a reduction in the root cortex, shortening the distance between the epidermis and the mid-column, thus facilitating the uptake of essential minerals from the soil [35].” If this salt stress-related anatomy change appears it would be advantageous due to enhancing effect on the short-distance ion transport into the stele vascular tissue.

Response 19: The reference here is not properly quoted. We have deleted it. The opinions of the reviewers are very pertinent and provide ideas for our later research, and we will slice and observe the internal structure of plants to explore this possibility.

Point 20. L373 Please correct this expression: plasma wall separation in plant cells to e.g. plasma-cell wall separation or plasma membrane-cell wall separation.

Response 20: We have made correction according to the Reviewer’s comments. The revised details can be found in Line 399, page 13.

Point 21. L446-448 I do not fully agree with this statement since You measured K only in roots but it is also presumed that decrease in K concentration in the roots might be attributed to the enhanced K transport into the shoots (leaves) under NaCl treatment.

Response 21: The distribution of sodium and potassium content in plant roots, stems and leaves is indeed worthy of further discussion, but our research results do find that inoculation of JZ-GX1 strain can effectively increase potassium ions in roots and increase the ratio of sodium to potassium, thus alleviating the salt stress of camphor. The revised details can be found in Line 472-473, page 14.

Point 22. L465 Modelling and better understanding the potential influence of PGPR on the whole plant physiology in salty environment would require more information on the change in ion homeostasis in leaves too.

Response 22: We think that the root system of the plant is first exposed to sodium chloride, which may be more targeted than the leaf, so the ion content in the leaf is not determined.

Reviewer 2 Report

The manuscript is generally well written and has enough clarity. The study was undertaken when the plants were at the seedling stage and yet C. camphora is a tree with a long growth period. How are the authors extrapolating the results they have obtained to the full-grown tree? This point should be included in the discussion.

About the methods: The soil used to grow the plants: it seems it was collected from somewhere. Is that correct? Soil composition may vary according to site of collection. Not clear how the authors standardized the soil.

Also how was the final salinity of the soil maintained and measured? More details are required. 

It has not been mentioned how the bacteria were applied. To the seeds? to germinated seedlings or to the soil?

For the root study, how did the investigators ensure that roots remained intact, despite being pulled out from soil?

Author Response

Response to Reviewer 2 Comments

Point 1. The manuscript is generally well written and has enough clarity. The study was undertaken when the plants were at the seedling stage and yet C. camphora is a tree with a long growth period. How are the authors extrapolating the results they have obtained to the full-grown tree? This point should be included in the discussion.

Response 1: In our study, JZ-GX1 improves the salt tolerance of camphor seedlings mainly by improving antioxidant capacity and regulating ion balance, which we speculate to be achieved by secondary metabolites secreted by inoculants. However, as the reviewers said, it is not clear whether this effect can be reproduced in big trees, but at least our research shows that JZ-GX1 has the potential to alleviate camphor salt stress. This effect may vary depending on the size of the seedlings and the planting area. The revised details can be found in Line 487-489, pages 14-15.

Point 2. About the methods: The soil used to grow the plants: it seems it was collected from somewhere. Is that correct? Soil composition may vary according to site of collection. Not clear how the authors standardized the soil.

Response 2: The soil we use to cultivate plants is collected from one place, and the quality of the soil in each basin is consistent in order to ensure the standard of the experiment.

Point 3. Also how was the final salinity of the soil maintained and measured? More details are required.

Response 3: According to 1%=170 mM, we convert that 50 mM is about 0.3%, that is, 0.3 g of NaCl is added to 100 g soil, of which 0.3 g of sodium chloride is dissolved in 50 mL distilled water, the concentration of 100 mM, and so on. The revised details can be found in Line 105-117, page 3.

Point 4. It has not been mentioned how the bacteria were applied. To the seeds? to germinated seedlings or to the soil?

Response 4: We collected 3-month-old camphor seedlings from the nursery, then transplanted them into flowerpots, and then inoculated bacteria, followed by salt stress treatment.

Point 5. For the root study, how did the investigators ensure that roots remained intact, despite being pulled out from soil?

Response 5: The soil used in our experiment is loose and often watered, so the seedlings are easy to take out of it.